# Diffusional Interactions among Marine Phytoplankton and Bacterioplankton: Modelling H_2_O_2_ as a Case Study

**DOI:** 10.3390/microorganisms10040821

**Published:** 2022-04-15

**Authors:** Naaman M. Omar, Ondřej Prášil, J. Scott P. McCain, Douglas A. Campbell

**Affiliations:** 1Department of Biology, Mount Allison University, Sackville, NB E4L1G7, Canada; nomar@mta.ca; 2Center Algatech, Laboratory of Photosynthesis, Novohradska 237, CZ 37981 Trebon, Czech Republic; prasil@alga.cz; 3Department of Biology, Massachusetts Institute of Technology, Boston, MA 02142, USA; smccain@mit.edu

**Keywords:** diffusional interactions, hydrogen peroxide, phytoplankton, bacterioplankton

## Abstract

Marine phytoplankton vary widely in size across taxa, and in cell suspension densities across habitats and growth states. Cell suspension density and total biovolume determine the bulk influence of a phytoplankton community upon its environment. Cell suspension density also determines the intercellular spacings separating phytoplankton cells from each other, or from co-occurring bacterioplankton. Intercellular spacing then determines the mean diffusion paths for exchanges of solutes among co-occurring cells. Marine phytoplankton and bacterioplankton both produce and scavenge reactive oxygen species (ROS), to maintain intracellular ROS homeostasis to support their cellular processes, while limiting damaging reactions. Among ROS, hydrogen peroxide (H_2_O_2_) has relatively low reactivity, long intracellular and extracellular lifetimes, and readily crosses cell membranes. Our objective was to quantify how cells can influence other cells via diffusional interactions, using H_2_O_2_ as a case study. To visualize and constrain potentials for cell-to-cell exchanges of H_2_O_2_, we simulated the decrease of [H_2_O_2_] outwards from representative phytoplankton taxa maintaining internal [H_2_O_2_] above representative seawater [H_2_O_2_]. [H_2_O_2_] gradients outwards from static cell surfaces were dominated by volumetric dilution, with only a negligible influence from decay. The simulated [H_2_O_2_] fell to background [H_2_O_2_] within ~3.1 µm from a *Prochlorococcus* cell surface, but extended outwards 90 µm from a diatom cell surface. More rapid decays of other, less stable ROS, would lower these threshold distances. Bacterioplankton lowered simulated local [H_2_O_2_] below background only out to 1._2_ µm from the surface of a static cell, even though bacterioplankton collectively act to influence seawater ROS. These small diffusional spheres around cells mean that direct cell-to-cell exchange of H_2_O_2_ is unlikely in oligotrophic habits with widely spaced, small cells; moderate in eutrophic habits with shorter cell-to-cell spacing; but extensive within phytoplankton colonies.

## 1. Introduction

### 1.1. Marine Phytoplankton

Phytoplankton are a polyphyletic functional grouping of ~275,000 species of oxygenic photosynthetic microorganisms that grow while suspended in marine or freshwater habitats [1,2]. Phytoplankton interact with their aqueous environment [3], and with co-occurring bacterioplankton cells, through diffusion of nutrients, toxic compounds and signaling molecules, notably including reactive oxygen species (ROS). Phytoplankton span a wide size range [4] (Table 1), which directly affects their diffusional interactions. Larger cells have longer paths for intracellular diffusion or transport paths for solutes, and can therefore potentially impose larger concentration gradients between their internal cell contents and the local microenvironment. Larger cells also have lower surface to volume ratios which impose limits on their trans-membrane transport per biovolume of the cell [5]. On the other hand, some larger cells have dynamic control over their sinking through water, and can thereby refresh their surface boundary layers more effectively than can smaller cells [3,6].

### 1.2. Phytoplankton Cell Spacing in Oceans

Phytoplankton and bacterioplankton cell numbers in the ocean vary widely, and therefore so does intercellular distance (Table 1). Direct colonization of phytoplankton by bacteria [7], or colony-forming phytoplankters, bring multiple cells into yet closer local proximities. For example, within *Phaeocystis* colonies, cell densities reach 2.4 × 10^7^ cells mL^−1^ [8] (Table 1).

**Table 1 microorganisms-10-00821-t001:** Phytoplankton and bacterioplankton cell sizes and representative suspension densities.

Habitat	Taxa	Metabolism	Cell m^−3^	Cell Radius µm	Citation
Oligotrophic	Bacterioplankton	Hetero	4.7 × 10^10^	0.5	[9]
Oligotrophic	Bacterioplankton	Hetero	2.0 × 10^11^	0.5	[10]
Oligotrophic	Diatoms	Phyto	5.6 × 10^7^	10.0	[11]
Oligotrophic	Dinoflagellates	Phyto	6.5 × 10^6^	10.0	[11]
Oligotrophic	PicoEukaryotes	Phyto	6.7 × 10^8^	2.0	[12]
Oligotrophic	PicoEukaryotes	Phyto	2.0 × 10^9^	2.0	[10]
Oligotrophic	Prochlorococcus	Phyto	4.7 × 10^10^	0.3	[12]
Oligotrophic	Prochlorococcus	Phyto	3.8 × 10^10^	0.3	[10]
Oligotrophic	Silicoflagellates	Phyto	1.3 × 10^7^	10.0	[11]
Oligotrophic	Synechococcus	Phyto	3.0 × 10^9^	1.0	[12]
Oligotrophic	Synechococcus	Phyto	2.0 × 10^9^	1.0	[10]
Mesotrophic	Bacterioplankton	Hetero	3.3 × 10^11^	0.5	[10]
Mesotrophic	PicoEukaryotes	Phyto	3.0 × 10^9^	2.0	[10]
Mesotrophic	Prochlorococcus	Phyto	3.1 × 10^10^	0.3	[10]
Mesotrophic	Synechococcus	Phyto	6.0 × 10^9^	1.0	[10]
Eutrophic	Bacterioplankton	Hetero	4.6 × 10^11^	0.5	[10]
Eutrophic	Bacterioplankton	Hetero	4.6 × 10^13^	0.5	[13]
Eutrophic	Chlorophytes	Phyto	8.2 × 10^10^	2.5	[14]
Eutrophic	Chroococcales	Phyto	8.2 × 10^10^	1.0	[14]
Eutrophic	Cyanobium	Phyto	4.0 × 10^12^	1.0	[14]
Eutrophic	PicoEukaryotes	Phyto	3.0 × 10^9^	2.0	[10]
Eutrophic	PicoEukaryotes	Phyto	1.5 × 10^13^	2.0	[13]
Eutrophic	Synechococcus	Phyto	4.5 × 10^9^	1.0	[10]
Colony	Cyanobium	Phyto	1.0 × 10^13^	1.0	[14]
Colony	Phaeocystis	Phyto	2.4 × 10^13^	2.2	[8]

These wide ranges of cell suspension densities for phytoplankton and bacterioplankton across habitats strongly influence the extent to which the growth and activity of a phytoplankton population or community alters, or even governs, solute concentrations in the local habitat [15,16,17,18]. Beyond the bulk activities of the phytoplankton population or community, differences in cell suspension densities mean the diffusional path lengths for solutes separating individual cells vary by orders of magnitude across habitats and taxa. Intercellular distances then influence whether the activity of an individual cell can directly influence the microenvironment of neighboring cells [19].

Our understanding of the taxa- and habitat-specific effects of cell-to-cell separation is then influenced by schematic visualizations of phytoplankton communities with implied cell suspension densities far higher than is realistic, or with cell symbol sizes scaled differently than the scaling of the spatial axes [20]. Symbol size scaling exaggerated relative to the scaling of the spatial axes allows schematic visualizations of cells, but can give a visual impression of a community with fairly close cell-to-cell interactions. In our visualizations, we maintain cell symbol size at the same size scaling as the spatial axes (Figure 1).

### 1.3. Reactive Oxygen Species

Electrons can transfer from photosynthetic or respiratory electron transport to reduce O_2_ to form the superoxide radical, a reactive oxygen species (ROS) [21,22] (Reaction (1)).

ROS include both radicals (superoxide, nitric oxide and hydroxyl radicals) and non-radicals (hydrogen peroxide, singlet oxygen and peroxynitrite), which are more reactive than ground state di-oxygen molecules [23,24]. Reduction of O_2_ to H_2_O requires the sequential addition of four electrons, and therefore progresses through three ROS intermediates: superoxide (O_2_^•−^), hydrogen peroxide (H_2_O_2_) and hydroxyl radical (OH^•^) (Reaction (1)) [25]. ROS are by-products of both photosynthetic and heterotrophic metabolism, but are also generated abiotically in seawater [26].
(1)$$\begin{array}{c} O_{2}+e^{-}\to O_{2}^{•-}\\ O_{2}^{•-}+2H^{+}+e^{-}\to H_{2}O_{2}\\ H_{2}O_{2}+H^{+}+e^{-}\to H_{2}O+{\mathrm{OH}}^{•}\\ {\mathrm{OH}}^{•}+e^{-}+H^{+}\to H_{2}O \end{array}$$

Biogenic production of extracellular ROS mediated by microorganisms is a significant contributor to marine environments [27,28], mediated by taxa ranging from heterotrophic bacteria to diatoms including those from the *Thalassiosira* and *Coscinodiscus* genera [28,29,30,31,32,33,34,35].

ROS are important at low concentrations due to their role in cell signaling [36,37,38,39,40,41,42] and lipid accumulation [43]. In order to benefit from ROS, organisms must maintain ROS levels above the cytostatic threshold, the minimum concentration of ROS required to maintain normal cellular processes, as well as below the cytotoxic threshold, the level at which the negative effects of ROS outweigh the positives [44].

### 1.4. Concentrations and Properties of H_2_O_2_

H_2_O_2_ is an uncharged compound documented in the ocean at concentrations of 10^−9^ to 10^−6^ mol L^−1^ [29,45] depending upon location and conditions. These [H_2_O_2_] are high enough to significantly contribute to the redox cycling of copper [46,47]. [H_2_O_2_] in seawater increases after precipitation, including rain and snowfall [18,26,48]. [H_2_O_2_] in seawater also follows a diurnal cycle with a peak at mid-day [18,49,50], which suggests significant direct or indirect photochemical or photobiological generation of H_2_O_2_. [H_2_O_2_] also exhibits latitudinal variation at the surface, higher in the brighter, warmer, mid to low latitudes compared to the colder, darker, higher latitudes [50]. The authors of [50] suggest that regional variation in [H_2_O_2_] may be caused by the depth of the mixed layer and the concentration of total dissolved organic carbon. Abiotic production rates of H_2_O_2_ vary from 0.28 to 4.2 pM s^−1^ in near surface open ocean waters vs. 1.0 to 84.4 pM s^−1^ in coastal sites [26].

The diffusion coefficients (a measure of how quickly a molecule travels via diffusion), and therefore the diffusional mobilities, of ROS through seawater vary widely. Diffusion coefficients in water decrease with increasing molecular mass, and from uncharged to charged compounds. Diffusion coefficients are strongly influenced by temperature and slightly influenced by pressure. For H_2_O_2_ at ~20 °C the diffusion coefficient is ~1500 µm^2^ µs^−1^ (note conversion of units from more typical reporting of cm^2^ s^−1^) [51]. In aquatic systems, H_2_O_2_ has extracellular lifetimes of hours to days [29,52] before decay.

H_2_O_2_ is acutely toxic to most cells in the range of 10^−5^ to 10^−4^ mol L^−1^ range [53], about 10-fold higher than seawater concentrations [29,45]. We found few quantitative estimates of the intracellular concentrations of H_2_O_2_ within cells [54,55,56,57,58,59,60], with a particular lack of measurements from phytoplankton. Limited data, from mammalian cells, support an intracellular H_2_O_2_ concentration of 10^−6^ M [61]. Although only weakly directly reactive, H_2_O_2_ can react with thiols and methionine [62] and thereby modulate gene expression and transcription [63,64]. Meanwhile, limited experimental data indicate that heterotrophic bacteria maintain an internal [H_2_O_2_] of 1/10 of the external [H_2_O_2_] when extracellular H_2_O_2_ is present [58,60], as it is in seawater. The authors of [58] also predicted an intracellular steady-state [H_2_O_2_] of 20 nM in the absence of extracellular H_2_O_2_, which is far below the toxic threshold of intracellular H_2_O_2_ [60], and is likely for the maintenance of physiological roles of H_2_O_2_. Cytotoxic effects of H_2_O_2_, including lipid damage, are primarily caused by H_2_O_2_ converting to the hydroxyl radical, which is strongly oxidative [52]. Interestingly, H_2_O_2_ may be used as a terminal electron acceptor in the absence of oxygen in *Escherichia coli* [65].

H_2_O_2_ readily crosses the cell membrane [53] primarily through aquaporins [66,67,68,69]. As a result, intracellular H_2_O_2_ exchanges with extracellular H_2_O_2_, but cells can nevertheless maintain intracellular [H_2_O_2_] significantly different than extracellular [H_2_O_2_] [58]. Other ROS rarely cross cell membranes and therefore the intra- and extracellular pools are not directly connected, although they may influence each other indirectly through transformations to other ROS, for example dismutation of intracellular O_2_^•-^ to form intracellular H_2_O_2_ which can then exchange with extracellular H_2_O_2_, and vice versa.

A significant fraction of H_2_O_2_ destruction in aqueous systems may be attributed to the activities of heterotrophic microbes [70], although heterotrophs are minor contributors to bulk [H_2_O_2_]. Microbial scavenging of H_2_O_2_ indeed increases the survival of *Prochlorococcus* through extended periods of darkness, as demonstrated through co-culture studies [71] with ‘helper’ bacteria which carry genes for catalase. In the open ocean, bulk [H_2_O_2_] levels generally remain below the cytotoxic threshold, partly because heterotrophic organisms provide catalase [72].

Changes in [H_2_O_2_], or other ROS, around a cell will depend upon net production, or consumption by the cell, which generate net release (or uptake) of [H_2_O_2_] from the cell after any intracellular or cell-associated scavenging. These inputs can interact to generate a diffusion-generated gradient of [H_2_O_2_] around the cell, above or below seawater [H_2_O_2_]. Whether or not a cell establishes a zone of [H_2_O_2_] different from the [H_2_O_2_] background also depends upon extracellular destruction and diffusion rates. The extent of this altered zone will then influence downstream processes. For specific cell–cell interactions or responses, a cell has to change the [H_2_O_2_] in the local microenvironment sufficiently to provoke a response in neighboring cells. Beyond this local sphere of influence around an individual cell, the collective activity of cells in a community contributes to generating a general seawater [H_2_O_2_], which in turn influences cellular gene expression and metabolism.

### 1.5. Goals

Microbes consume and produce H_2_O_2_, thereby potentially helping, or at least influencing, other microbes. We sought to quantify the influences of cell density and cell size on these potential microbial interactions. H_2_O_2_ presents a good case study for diffusional interactions among microbes. H_2_O_2_ readily traverses cell membranes, and there is information on background concentrations of H_2_O_2_ in seawater, H_2_O_2_ diffusion coefficients in seawater and some estimates of cellular concentrations of H_2_O_2_, as either absolute values or ratios compared to external [H_2_O_2_]. We can therefore approximate concentration gradients generated by [H_2_O_2_] diluting outwards from a cell maintaining a homeostatic intracellular [H_2_O_2_] higher than seawater [H_2_O_2_], or the concentration gradient inwards towards a hypothetical cell maintaining a homeostatic intracellular [H_2_O_2_] lower than seawater [H_2_O_2_]. We can use these simple approximations to generate quantitative thresholds for when H_2_O_2_ released from a cell can directly influence neighboring cells, i.e., how much, and how far, a phytoplankton or bacterioplankton cell can change the local [H_2_O_2_].

## 2. Results and Discussion

### 2.1. Cell to Cell Spacing across Habitats and Taxa

Possible cell-to-cell exchanges of H_2_O_2_ are mediated by diffusions through intercellular path lengths, varying by orders of magnitude across habitats and taxa. To understand such diffusional exchanges we need realistic visualizations of cell spacings, achieved by maintaining cell symbol size at the same size scaling as the spatial axes (Figure 1). In the visualization of randomized cell spacing under oligotrophic conditions, we see wide spacing among cells, because the small cell symbols are on the same size scaling as the spatial axes. In the visualization of randomized cell spacing under eutrophic conditions, some cells are in close proximity. Note that this standardized scaling visualization approach is only feasible with highly expanded spatial axes such as the 1 mm axis used in Figure 1, equivalent to simulation of only 1 µL. Attempting to visualize 10 mm axes, equivalent to a 1 mL volume, causes the cell symbol size to shrink below visibility (data not presented). A further nuance not captured in these visualizations and subsequent simulations is non-random clustering of cells, as perhaps around particles of marine snow.

### 2.2. H_2_O_2_ Concentration Gradients around Cells

Figure 2 shows a light blue background for the approximate range of natural [H_2_O_2_] in seawater, with the simulated profiles of [H_2_O_2_] moving outward from a ‘Bacterioplankton’ cell with intracellular [H_2_O_2_] maintained below seawater [H_2_O_2_]; or outward from small phytoplankton cells of different sizes with intracellular [H_2_O_2_] maintained at 1 µmol L^−1^ [57], near the highest concentrations expected for seawater [ROS]. For [H_2_O_2_] generators (nominally, phytoplankton of different sizes), [H_2_O_2_] concentrations are highest near the cell surface and decrease with distance outwards. For [H_2_O_2_] consumers (nominally, bacterioplankton), [H_2_O_2_] concentrations are lowest near the cell surface.

Decay is a key influence on the pseudo-steady state bulk [H_2_O_2_] in seawater, but for our conceptual estimates of [H_2_O_2_] changing within the local region of the cell, the dilution term (black points, Figure 2) dominates the concentration gradient moving out from the cell. Including the influence of the decay term (red points, Figure 2) has only a negligible influence on [H_2_O_2_] moving out from the phytoplankton cell over simulated timescales of <100 µs and distances of ~100 µm. For other, less stable ROS, decay would become a significant influence, acting to narrow the local concentration gradients around cells.

Seawater contains H_2_O_2_ which varies significantly depending upon conditions and region. At some threshold distance out from the cell-specific sphere of influence upon local [ROS], it reaches the seawater [H_2_O_2_] (Figure 2), beyond which distance net [H_2_O_2_] production or consumption by the specific cell has no local influence on [H_2_O_2_], but is simply one input into the wider seawater [H_2_O_2_]. All other factors being equal, in our estimations the region over which a cell changes the local [H_2_O_2_] away from seawater [H_2_O_2_] increases with cell radius. This wider influence of cells with larger cell radii results primarily from the volume of the intracellular homeostatic [H_2_O_2_] increasing as the cube of cellular radius (Figure 2).

In this simple estimate, a heterotrophic bacterioplankton cell maintaining intracellular [H_2_O_2_] at 1/10 of the external seawater [H_2_O_2_] locally drops [H_2_O_2_] below the lower end of the seawater [H_2_O_2_] for a radius of only 1.2 µm. If the bacterioplankton cell were to maintain an intracellular [H_2_O_2_] of <1/10 of the external seawater [H_2_O_2_], the threshold distances would be greater. The phytoplankton cells maintaining intracellular [H_2_O_2_] at 1 µmol L^−1^ influence local [H_2_O_2_] above a low background level of seawater [H_2_O_2_] for radii of 3.1 µm for *Prochlorococcus* up to 90 µm for diatoms, or even more for yet larger phytoplankters not simulated. Furthermore, these threshold distances apply to the lowest published estimates for seawater [H_2_O_2_]. Under environments with higher seawater [H_2_O_2_], the threshold distances would be smaller.

### 2.3. Visualizing Cell to Cell Exchange of [H_2_O_2_]

We next combine our visualizations of cell suspensions from eutrophic, oligotrophic, or colonial habitats, overlaid with the mathematical expressions of cell-specific spheres of [H_2_O_2_] above or below seawater levels, to visualize the extent to which cells directly influence the [H_2_O_2_] environments of their neighbors.

In eutrophic habitats, bacterioplankters are numerous, but their small cell size, and correspondingly small cell-specific spheres of upon local [H_2_O_2_], mean that their individual contributions towards local [H_2_O_2_] are negligible (Figure 3), even though their collective influence upon background [H_2_O_2_] may be considerable [71,72]. Phytoplankton cells, particularly the larger taxa, can however influence local [H_2_O_2_] over wider distances, leading to overlaps in their cell-specific spheres of influence (Figure 3) with bacterioplankton and with other phytoplankton, and opening the possibility of direct cell-to-cell signaling or cell-to-cell metabolic influences through [H_2_O_2_], beyond contributions to background seawater [H_2_O_2_]. In oligotrophic habitats, wider cell-to-cell spacings, and generally smaller cell sizes, make direct cell-to-cell reciprocal influences upon local [H_2_O_2_] unlikely, although the phytoplankton and bacterioplankton as a whole indeed contribute to establishment of the background [H_2_O_2_].

For phytoplankters growing colonially, the [H_2_O_2_] leaving specific cells within the colony strongly influence the local [H_2_O_2_] environments of neighboring cells in the colony (Figure 4). Such a colony could also be usefully simulated as a single homeostatic sphere of [H_2_O_2_], with a significant local sphere of influence beyond the colony. Although we did not include bacteria in Figure 4, in fact *Phaeocystis antarctica* colonies have rich bacterial components [73], which may greatly alter the local H_2_O_2_ environment within the colony, beyond the activities of the *Phaeocystis* cells.

### 2.4. Threshold Distances for Cell to Cell Exchange of [H_2_O_2_]

We find that cell size, cell suspension density, cellular [H_2_O_2_] and seawater [H_2_O_2_] interact to influence the potential for direct cell-to-cell diffusional interactions. We sought to generalize the estimations presented earlier for combinations of taxa x habitat (Figure 3) to summarize how these variables potentially affect cell-to-cell interactions via H_2_O_2_ (Figure 5).

Figure 5 plots combinations of cell suspension density (X axis, governing cell-to-cell distance), and the radii of the cell-specific sphere of influence upon [H_2_O_2_] exceeding seawater [H_2_O_2_] (Y axis), for homogeneous populations. The threshold lines mark the transition for cell-to-cell interactions, for four different ratios of cellular [H_2_O_2_] to seawater [H_2_O_2_]. The areas above the threshold lines show the regions of cell radius and cell suspension density where the cell-specific spheres of influence upon [H_2_O_2_] exceed the average cell-to-cell spacing, for a given ratio of cellular [H_2_O_2_] to seawater [H_2_O_2_]. For the purpose of presentation, we collapsed cellular [H_2_O_2_] and seawater [H_2_O_2_] to the ratio of cellular [H_2_O_2_] to seawater [H_2_O_2_] since that ratio determines the position of the threshold line, rather than the absolute values for cellular [H_2_O_2_] and seawater [H_2_O_2_]. We see that a homogeneous population of large cells of 40 µm radius would maintain overlapping cell-specific spheres of influence down to cell suspension densities of 2.7 × 10^9^ cells m^3^ (Figure 5). In contrast, a homogeneous population of cells of 0.5 µm radius would only achieve directly interacting cell-specific spheres of influence at a hypothetical 1.9 × 10^15^ cells m^3^, far above cell suspension densities for most marine habitats (Table 1). For comparison, in Figure 5 we overlay symbols for representative taxa from different habitats (Table 1) showing that for hypothetical homogeneous populations the cell-specific spheres of influence are generally far below the expected cell-to-cell spacing.

Figure 6 shows that for natural population densities of bacterioplankton (Figure 6; Points), the spheres of lowered local [H_2_O_2_] would overlap when [H_2_O_2_] intracellular:extracellular ratio is less than or equal to 0.01 (Figure 6, short dashed line). However, the limited available information [58] suggests that heterotrophic bacteria maintain an [H_2_O_2_] intracellular:extracellular ratio of about 0.1 (Figure 6, dotted line). It is possible that cells may not be able to maintain an intracellular:extracellular [H_2_O_2_] ratio of 0.1 at the lowest seawater [H_2_O_2_] given the likely roles of H_2_O_2_ in cell signaling [36], and the need to maintain a basal [H_2_O_2_] [44]. The authors of [58] predict an intracellular steady-state [H_2_O_2_] of 20 nM in heterotrophs in the absence of extracellular H_2_O_2_, which would suggest that at low extracellular [H_2_O_2_], the intracellular:extracellular [H_2_O_2_] ratio would be >0.1, and the spheres of lowered local [H_2_O_2_] would not overlap (Figure 6). Thus, the cell-specific effects of bacterioplankton upon local [H_2_O_2_] are unlikely to overlap at reasonable cell suspension densities.

## 3. Conclusions

Our approximations of cell-to-cell spacing are applied to each taxa x habitat combination separately, based upon published measures of cell suspension densities in marine systems. We then overlaid the taxa-specific simulations from a given habitat to give a visualization of community level cell-to-cell spacings, across taxa within a habitat (Figure 1). As a next step we aim to implement probability distributions of cell-to-cell spacings across all cells from a mixed community in the habitat, to more accurately capture the likelihood of cell-to-cell interactions. Our work used available cell suspension density data exclusively from marine habitats. In general, we would assume that if cell densities and [H_2_O_2_] in freshwater systems are similar to that of marine systems, the diffusional patterns described for marine habitats would be comparable to freshwater systems, but future studies would need to confirm this.

Our simple approximations of diffusion gradients out from cells are limited by a dearth of published information on the intracellular concentrations of H_2_O_2_, and other ROS, in phytoplankton or bacterioplankton cells. There are published estimates of intracellular and extracellular H_2_O_2_ production and decay rates in seawater, which could support more sophisticated reaction/diffusion simulation approaches.

The threshold lines for interacting cell-specific spheres of influence as functions of cell radii and cell suspension densities are generated for individual ‘taxa’ from marine environments with uniform intracellular [H_2_O_2_] (Figure 5), rather than for a mixed community of different cell sizes and physiologies. These fixed threshold lines also apply to a fixed ratio of internal to external [H_2_O_2_], whereas this ratio could vary rapidly with changes in cell physiology and external conditions, causing the threshold for direct cell-to-cell interactions to shift. Our models suggest that it is unlikely that H_2_O_2_ diffusing out of a phytoplankton cell directly interacts with another phytoplankton cell at natural seawater cell densities (Figure 5), with the exception of *Phaeocystis*, whereby H_2_O_2_ diffusing out of a cell would directly interact with other cells if the intracellular:extracellular [H_2_O_2_] ratio is greater than 100. A future approach aims to generate probability distributions for direct cell-to-cell interactions, since even populations with widely spaced cells occasionally generate close cell-to-cell distances. Analyses show higher rates of cell-normalized H_2_O_2_ production in bloom-forming phytoplankton taxa, providing support for the selective influence of cell suspension density upon ROS dynamics [29].

It is likely that extracellular ROS play a role in the growth and development of phytoplankton, given that [74] found that the addition of superoxide dismutase and catalase inhibited the growth of *Chattonella marina*. Our models suggest that H_2_O_2_ diffusing out of phytoplankton cells reaches external [H_2_O_2_] before interacting with other cells (Figure 3 and Figure 5), and are primarily driven by the influence of diffusion, and with negligible contribution from the decay of H_2_O_2_ (Figure 2). Indirect interaction of H_2_O_2_ diffusing out of cells is also possible as H_2_O_2_ can affect biogeochemical cycles and bioavailability of nutrients [75].

The simple estimates presented herein using H_2_O_2_ as a case study are generalizable to other, less stable ROS where decay will be a more immediate influence. For example, O_2_^•−^ itself is not cell membrane permeable and so forms separate intra- and extracellular pools. However, O_2_^•−^ is metabolized to H_2_O_2_, both inside and outside cells, and so could indirectly influence [H_2_O_2_], and thereby alter cell-to-cell interactions beyond the simple diffusion-dominated simulations we present. For example, extracellular production of O_2_^•−^ by heterotrophic bacteria can significantly alter O_2_^•−^ concentrations in the dark ocean [28]. Heterotrophs might also produce enough O_2_^•−^ to indirectly influence concentrations of O_2_^•−^ in the photic zone of the ocean. The phytoplankter *T. weissflogii* [34] generates ranges of ~8 × 10^−16^ mol O_2_^•−^ cell^−1^ hr^−1^, while four raphidophytes show production ranges of 0.45 to 4 × 10^−12^ mol O_2_^•−^ cell^−1^ hr^−1^ [76], illustrating the wide range of extracellular production which can differentially influence extracellular [ROS] across communities or conditions.

The approaches presented here are a step towards visualizing and constraining estimates of cell-to-cell interactions in plankton communities. We next need to validate these approaches with quantitative estimates of intracellular [H_2_O_2_], and other [ROS], within phytoplankton and bacterioplankton cells.

## 4. Methods, Simplifying Assumptions, and Limitations

### 4.1. Estimation of Cell to Cell Spacing

A simplistic estimator of cell-to-cell spacing is the reciprocal of the cube root of the cell suspension density. For example, if cell suspension density is expressed as cells µL^−1^ (equivalent to cells mm^−3^), the reciprocal of the cube root gives a cell-to-cell spacing estimate in mm. This cube root estimate assumes equally spaced cells, which somewhat overestimates average cell-to-cell spacing, since equally spaced cells give a maximum spacing, not the average spacing of randomly located cells. An arithmetic correction based upon the gamma function of 4/3 generates a numeric correction to multiply the reciprocal of the cube root of the cell suspension density by 0.55 to approximate average spacing of cells [77] (Equation (2)).
(2)$$CellSpace\;\mathrm{mm}=\sqrt[{3}]{\frac{1}{Cell\;Count\;{µL}^{-1}}}*0.55$$

We generated simulated cell suspension densities for each taxa and habitat by generating points separated by the average cell spacing between cells, along each of the X, Y, Z spatial axes. We then added random variation to the values of the evenly spaced points (R jitter, factor = 2), to generate a XYZ coordinate cloud. In a parallel approach (data not presented) we directly generated normally distributed random distributions around the average cell spacings, but that approach was computationally slow for denser cell suspension densities. Overlaying the results from co-occurring taxa then gives a visualization of the community in a habitat (Figure 1). Although overlaying data from separately simulated taxa is adequate for visualization, a more sophisticated model would be required to explicitly generate and retrieve the distributions of cell-to-cell distances within, or across, taxa. Furthermore, even in oligotrophic habitats with low average cell suspension densities the few cells present may be clustered non-randomly, an ecological nuance not captured in our approach.

### 4.2. Simulating Concentration Gradients

To simulate H_2_O_2_ concentration gradients generated around a given cell, we consider the cell as a fixed spherical volume which maintains a steady, homeostatic intracellular [H_2_O_2_]. For simulation of phytoplankton cells, we set a homeostatic intracellular [H_2_O_2_] of 10^−6^ M, based upon limited literature from other eukaryotes [61], and as a level below the cytotoxic threshold of 10^−5^ M. For simulation of bacterioplankton cells, we set a homeostatic intracellular [H_2_O_2_] of 1/10 the external seawater [H_2_O_2_] [58,60].

Using the H_2_O_2_ diffusion co-efficient in water, and time increments from 0 µs with a log_10_ series upwards to 100 µs, we calculate the progressive net diffusional displacement of H_2_O_2_ outward from the cell surface. We then estimate dilution of [H_2_O_2_] with diffusion outwards from a source ‘Phytoplankton’ cell or diffusion inwards towards a sink ‘Heterotrophic’ cell. In addition to the simple volumetric dilution as ROS diffuses outwards from ‘Phytoplankton’ cells, we add a time-dependent decay term based upon measured lifetimes of H_2_O_2_ in seawater, which also vary widely. We use nominal time increments to generate the diffusion distances of H_2_O_2_ outward from the cell and to simulate the concurrent effect of pseudo-first order decay as H_2_O_2_ moves outward from the cell with time, but this simple simulation assumes a steady state, without considering local mixing or fluctuations in cellular or environmental H_2_O_2_.

### 4.3. Simulating Threshold Radii for [H_2_O_2_] above or below Seawater [H_2_O_2_]

On a visual plot, the intersect of the [H_2_O_2_] outwards from a ‘Phytoplankton’ source cell, with the lowest seawater [H_2_O_2_] is obvious, but it proved challenging to estimate this threshold distance when outwardly diffusing [H_2_O_2_] declines to the lowest seawater [H_2_O_2_]. The time for [H_2_O_2_] to decline to seawater [H_2_O_2_] depends upon the ratio of intracellular:extracellular [H_2_O_2_]; the diffusion coefficient for ROS, which determines how far [H_2_O_2_] moves out from the cell radius in a given time interval; and thus the [H_2_O_2_] volumetric dilution factor over a given time interval. For [H_2_O_2_], concomitant decay was only a negligible factor, but to retain capacity for generalizations to other, less stable [ROS] we nevertheless included an extracellular decay rate constant in the estimations. After algebraic transforms we found that a LambertW function was appropriate [78] to estimate the threshold radius at which [H_2_O_2_] around a cell falls to the lowest seawater [H_2_O_2_] (Equation (3)).

In Equation (3), [H_2_O_2_]_seawater_ is the seawater [H_2_O_2_] in the simulated habitat in mol L^−1^; [H_2_O_2_]_cell_ is the intracellular homeostatic [H_2_O_2_] in mol L^−1^; DiffusionRad_threshold_ in µm is the radius of the diffusion sphere when [H_2_O_2_] drops to [H_2_O_2_]_seawater_; CellRadius is the radius of the cell in µm; µ is the pseudo-first order decay constant for H_2_O_2_ in seawater; us_threshold_ is the time in µs at which [H_2_O_2_] drops to [H_2_O_2_]_seawater_; Diffusion Coefficient (D) is expressed in µm^2^ µs^−1^; W_0_ is the Lambert W function, also known as the Omega function, used to solve for f(x) = xe^x^.
(3)$$\begin{array}{c} {\left[H_{2}O_{2}\right]}_{\mathrm{seawater}}={\left[H_{2}O_{2}\right]}_{\mathrm{cell}}*\frac{\frac{4}{3}*\pi *{\left(\mathrm{CellRadius}\right)}^{3}}{\frac{4}{3}{*\pi *\mathrm{DiffusionRad}}_{\mathrm{threshold}}^{3}}{*e}^{-{\mu *\mathrm{us}}_{\mathrm{threshold}}}\\ {\left[H_{2}O_{2}\right]}_{\mathrm{seawater}}={\left[H_{2}O_{2}\right]}_{\mathrm{cell}}*\frac{{\left(\mathrm{CellRadius}\right)}^{3}}{{\mathrm{DiffusionRad}}_{\mathrm{threshold}}^{3}}{*e}^{-{\mu *\mathrm{us}}_{\mathrm{threshold}}}\\ {\left[H_{2}O_{2}\right]}_{\mathrm{seawater}}={\left[H_{2}O_{2}\right]}_{\mathrm{cell}}*\frac{{\left(\mathrm{CellRadius}\right)}^{3}}{{\left(\mathrm{CellRadius}+\sqrt[{2}]{{D*\mathrm{us}}_{\mathrm{threshold}}}\right)}^{3}}{*e}^{-{\mu *\mathrm{us}}_{\mathrm{threshold}}}\\ \frac{{\left[H_{2}O_{2}\right]}_{\mathrm{seawater}}}{{\left[H_{2}O_{2}\right]}_{\mathrm{cell}}*{\left(\mathrm{CellRadius}\right)}^{3}}=\frac{1}{{\left(\mathrm{CellRadius}+\sqrt[{2}]{{D*\mathrm{us}}_{\mathrm{threshold}}}\right)}^{3}}{*e}^{-{\mu *\mathrm{us}}_{\mathrm{threshold}}}\\ \frac{{\left[H_{2}O_{2}\right]}_{\mathrm{seawater}}}{{\left[H_{2}O_{2}\right]}_{\mathrm{cell}}*{\left(\mathrm{CellRadius}\right)}^{3}}=k\\ \sqrt[{3}]{k}=\frac{1}{\left(\mathrm{CellRadius}+\sqrt[{2}]{{D*\mathrm{us}}_{\mathrm{threshold}}}\right)}{*e}^{\frac{-{\mu *\mathrm{us}}_{\mathrm{threshold}}}{3}}\\ \mathrm{CellRadius}+\sqrt[{2}]{{D*\mathrm{us}}_{\mathrm{threshold}}}=\frac{e^{\frac{-{\mu *\mathrm{us}}_{\mathrm{threshold}}}{3}}}{\sqrt[{3}]{k}}\\ \sqrt[{2}]{{D*\mathrm{us}}_{\mathrm{threshold}}}=\frac{e^{\frac{-{\mu *\mathrm{us}}_{\mathrm{threshold}}}{3}}}{\sqrt[{3}]{k}}-\mathrm{CellRadius}\\ \frac{\sqrt[{2}]{{D*\mathrm{us}}_{\mathrm{threshold}}}}{e^{\frac{-{\mu *\mathrm{us}}_{\mathrm{threshold}}}{3}}}=\frac{1}{\sqrt[{3}]{k}}-\mathrm{CellRadius}\\ \frac{{D*\mathrm{us}}_{\mathrm{threshold}}}{e^{\frac{-{\mu *\mathrm{us}}_{\mathrm{threshold}}*2}{3}}}={\left(\frac{1}{\sqrt[{3}]{k}}-\mathrm{CellRadius}\right)}^{2}\\ {D*\mathrm{us}}_{\mathrm{threshold}}{*e}^{\frac{{\mu *\mathrm{us}}_{\mathrm{threshold}}*2}{3}}={\left(\frac{1}{\sqrt[{3}]{k}}-\mathrm{CellRadius}\right)}^{2}\\ {\mathrm{us}}_{\mathrm{threshold}}{*e}^{\frac{{\mu *\mathrm{us}}_{\mathrm{threshold}}*2}{3}}=\frac{{\left(\frac{1}{\sqrt[{3}]{k}}-\mathrm{CellRadius}\right)}^{2}}{D}\\ \frac{{\mu *\mathrm{us}}_{\mathrm{threshold}}*2}{3}=Y\\ \frac{Y}{\mu *\frac{2}{3}}{*e}^{Y}=\frac{{\left(\frac{1}{\sqrt[{3}]{k}}-\mathrm{CellRadius}\right)}^{2}}{D}\\ {Y*e}^{Y}=\frac{{\left(\frac{1}{\sqrt[{3}]{k}}-\mathrm{CellRadius}\right)}^{2}}{D}*\mu *\frac{2}{3}\\ Y=W_{0}\left(\frac{{\left(\frac{1}{\sqrt[{3}]{\frac{{\left[H_{2}O_{2}\right]}_{\mathrm{seawater}}}{{\left[H_{2}O_{2}\right]}_{\mathrm{cell}}*{\left(\mathrm{CellRadius}\right)}^{3}}}}-\mathrm{CellRadius}\right)}^{2}}{D}*\mu *\frac{2}{3}\right)\\ {\mathrm{us}}_{\mathrm{threshold}}=\frac{W_{0}\left(\frac{{\left(\frac{1}{\sqrt[{3}]{\frac{{\left[H_{2}O_{2}\right]}_{\mathrm{seawater}}}{{\left[H_{2}O_{2}\right]}_{\mathrm{cell}}*{\left(\mathrm{CellRadius}\right)}^{3}}}}-\mathrm{CellRadius}\right)}^{2}}{D}*\mu *\frac{2}{3}\right)}{\left(\mu *\frac{2}{3}\right)} \end{array}$$

For ‘Heterotrophic’ sink cells we simulated the sphere of [H_2_O_2_] below seawater [H_2_O_2_] accounting only for diffusion around the cell without including a rate constant for time-dependent bulk decay because the cell is acting, on top of any decay term, to further lower [H_2_O_2_] below seawater [H_2_O_2_] (Reaction (4)).
$${\left[H_{2}O_{2}\right]}_{\mathrm{seawater}}={\left[H_{2}O_{2}\right]}_{\mathrm{cell}}*\left({\mathrm{DiffusionSphere}}_{\mathrm{threshold}}\right)/\left(\left(4/3\right)*\pi *\mathrm{CellRadius}\right)$$
where [H_2_O_2_]_seawater_ is again the seawater [H_2_O_2_] in the simulated habitat in mol L^−1^; [H_2_O_2_]_cell_ is the intracellular homeostatic [H_2_O_2_] in mol L^−1^; and DiffusionRad_threshold_ in µm is the radius of the diffusion sphere when [H_2_O_2_] drops to [H_2_O_2_]_seawater_; CellRadius is the radius of the cell in µm.

Alter substitutions and re-arrangements:

Where µs_thresh is the elapsed time in µs to reach the threshold [H_2_O_2_]; Diffusion Coefficient (D) is expressed in µm^2^ µs^−1^; µs [1] is the first simulated time step. We then recover the threshold distance as:(4)$$\begin{array}{c} {\left[H_{2}O_{2}\right]}_{\mathrm{seawater}}={\left[H_{2}O_{2}\right]}_{\mathrm{cell}}*\frac{\left(4/3\right){*\pi *\mathrm{DiffusionRad}}_{\mathrm{threshold}}^{3}}{\left(4/3\right){*\pi *\mathrm{CellRadius}}^{3}}\\ {\left[H_{2}O_{2}\right]}_{\mathrm{seawater}}={\left[H_{2}O_{2}\right]}_{\mathrm{cell}}*\frac{{\mathrm{DiffuseRad}}_{\mathrm{threshold}}^{3}}{{\mathrm{CellRadius}}^{3}}\\ \frac{{\left[H_{2}O_{2}\right]}_{\mathrm{seawater}}}{{\left[H_{2}O_{2}\right]}_{\mathrm{cell}}}={\left(\frac{{\mathrm{DiffuseRad}}_{\mathrm{threshold}}}{\mathrm{CellRadius}}\right)}^{3}\\ {\mathrm{DiffuseRad}}_{\mathrm{threshold}}=\sqrt[{3}]{\frac{{\left[H_{2}O_{2}\right]}_{\mathrm{seawater}}}{{\left[H_{2}O_{2}\right]}_{\mathrm{cell}}}}*\mathrm{CellRadius} \end{array}$$

### 4.4. Simulations and Plotting

We used R [79] running under RStudio [80], using packages ‘tidyverse’ [81], ‘pracma’ [82], ‘ggplot2′ [83], ‘cowplot’ [84], ‘ggpubr’ [85], ‘glue’ [86], ‘kableExtra’ [87], ‘LambertW’ [88], ‘ggforce’ [89], ‘googledrive’ [90], and ‘googlesheets4’ [91]. Citations were managed using Zotero [92] open access reference manager connected to RStudio using the ‘citr’ [93] package, with output generated using the ‘knitr’ [94,95,96] and ‘bookdown’ [97] packages.

## Figures and Tables

**Figure 1 microorganisms-10-00821-f001:**
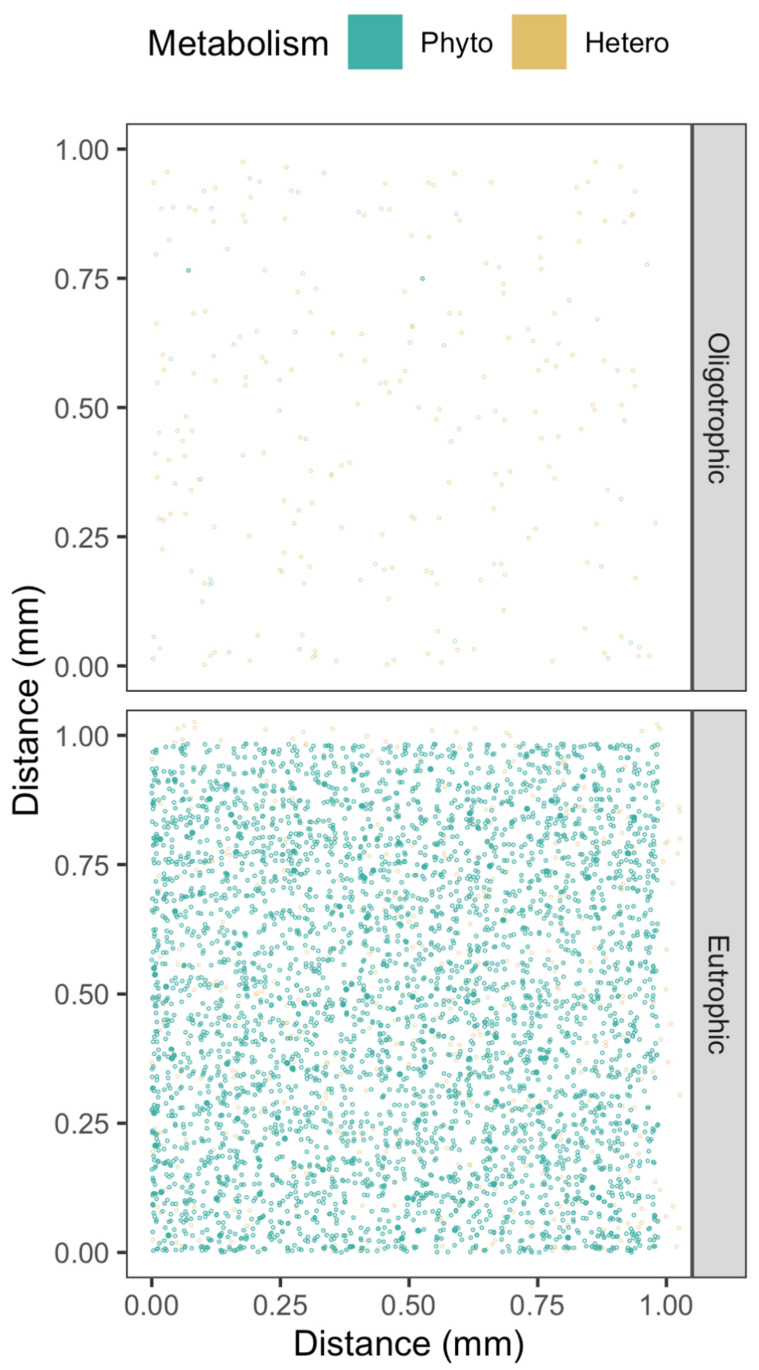
Phytoplankton and heterotrophic bacteria spacing from eutrophic or oligotrophic environments, with cell size on the same scaling as the 1 mm XY spatial axes. A 1 mm Z spatial axis is coded with fainter points farther back.

**Figure 2 microorganisms-10-00821-f002:**
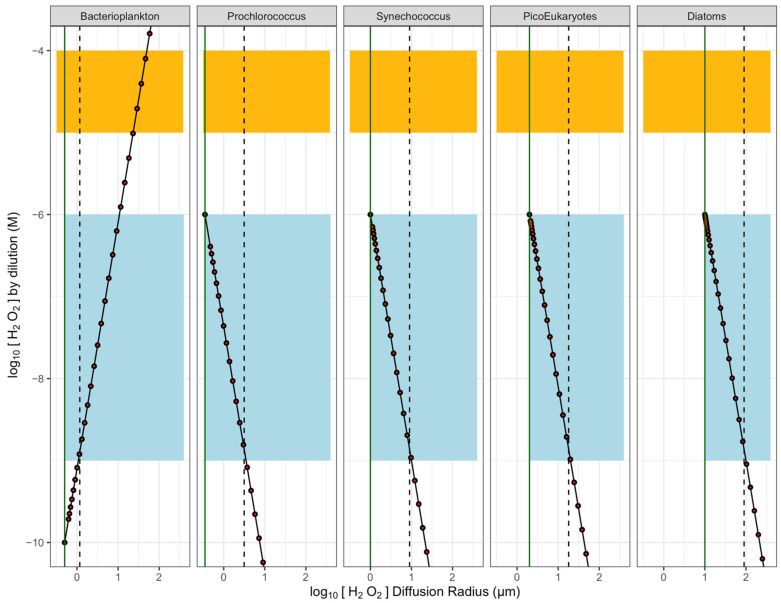
Estimates of concentration gradients of [H_2_O_2_] vs. distance outwards from ‘heterotrophic’ Bacterioplankton cell maintaining an internal [H_2_O_2_] below seawater background [H_2_O_2_] (blue band); or outwards from ‘Phytoplankton’ cells of different radii, maintaining an internal [H_2_O_2_] equal to 1/10 the cytotoxic threshold for [H_2_O_2_]. Orange indicates cytotoxic concentration range for [H_2_O_2_] (1 × 10^−5^ to 1 × 10^−4^ M). Black points show modelled [H_2_O_2_] under the influence of dilution alone. Red points show modelled [H_2_O_2_] under superimposed influences of dilution and (negligible) decay of H_2_O_2_. Vertical green lines indicate the cell surface. Vertical dashed lines indicate threshold radii out from cells, where [H_2_O_2_] around the cell falls or rises to seawater background levels. Beyond that threshold, the cell has no specific, local influence upon seawater [H_2_O_2_].

**Figure 3 microorganisms-10-00821-f003:**
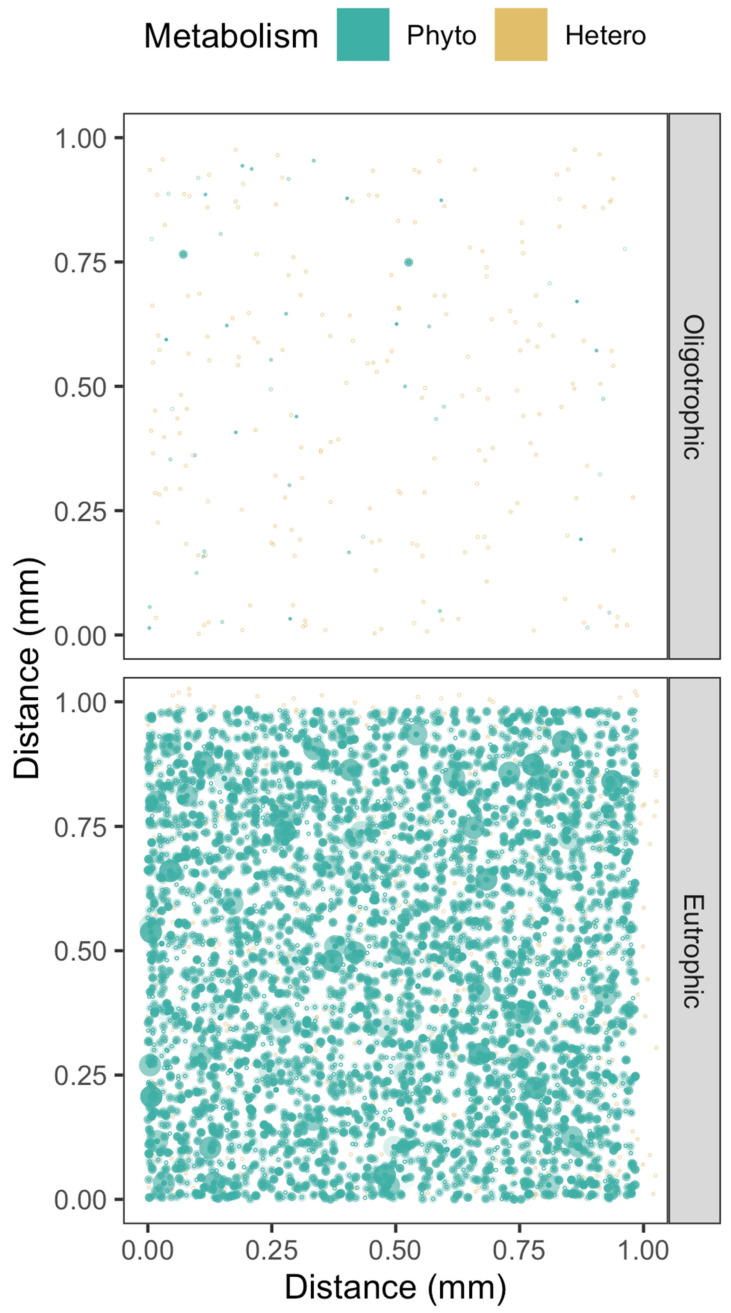
Simulations of [H_2_O_2_] above seawater [H_2_O_2_] (pale blue) out from Phytoplankton cells (green spheres, spanning a size range), or [H_2_O_2_] below seawater [H_2_O_2_] out from Bacterioplankton cells (red spheres, single nominal cell size). Cell position along a third Z axis is simulated by the ‘alpha’ with lighter cells further back.

**Figure 4 microorganisms-10-00821-f004:**
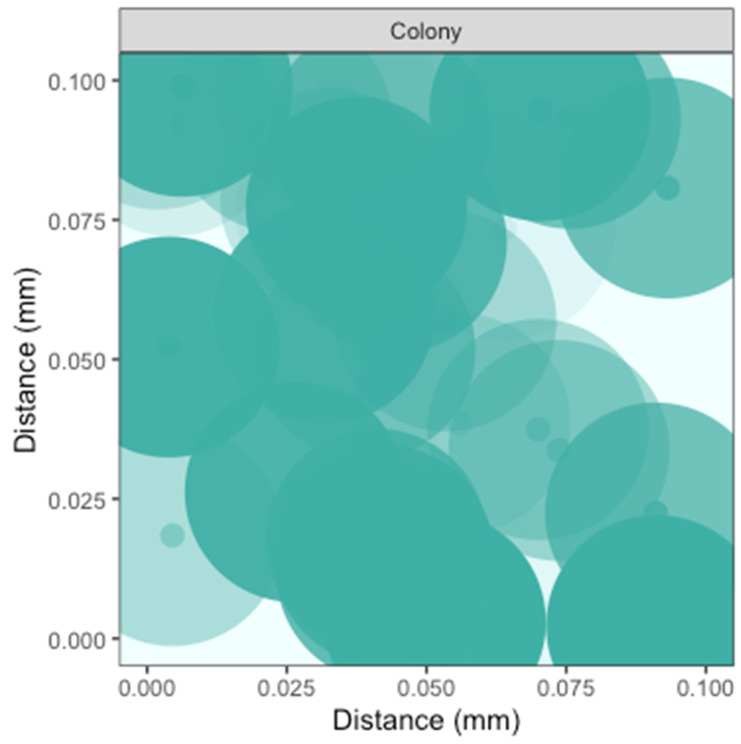
Simulation of [H_2_O_2_] above seawater [H_2_O_2_] within a Phaeocystis colony. Note the expansion of the axes scales to 0.1 mm. Cell position along a third Z axis is simulated by the ‘alpha’ with lighter cells further back.

**Figure 5 microorganisms-10-00821-f005:**
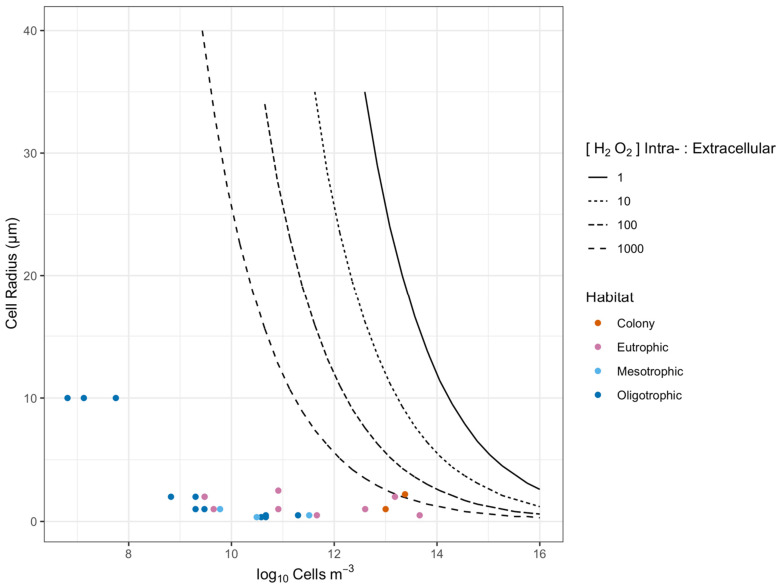
Thresholds for direct phytoplankton cell-to-cell interaction through [H_2_O_2_] above seawater background. The X axis shows a range of cell m^−3^ (which in turn governs cell-to-cell distances), while the Y axis shows radii for cell-specific spheres of influence upon [H_2_O_2_] (a function of cell size, intracellular [H_2_O_2_] and seawater [H_2_O_2_]). The areas below the threshold lines show combinations where phytoplankton cell-to-cell distance exceeds the cell-specific [H_2_O_2_] distance, so direct cell-to-cell interactions mediated by [H_2_O_2_] are unlikely. The area above the threshold lines show combinations where phytoplankton cell-to-cell distance is less than the cell-specific [H_2_O_2_] distance, so direct cell-to-cell interactions mediated by [H_2_O_2_] are feasible. The four threshold lines refer to a range of ratios of phytoplankton intracellular to extracellular [H_2_O_2_]. Data points from different habitats show that direct phytoplankton cell-to-cell interactions are likely only within the Phaeocystis colony, whereas single phytoplankton cells are generally too small or too distant for direct interactions.

**Figure 6 microorganisms-10-00821-f006:**
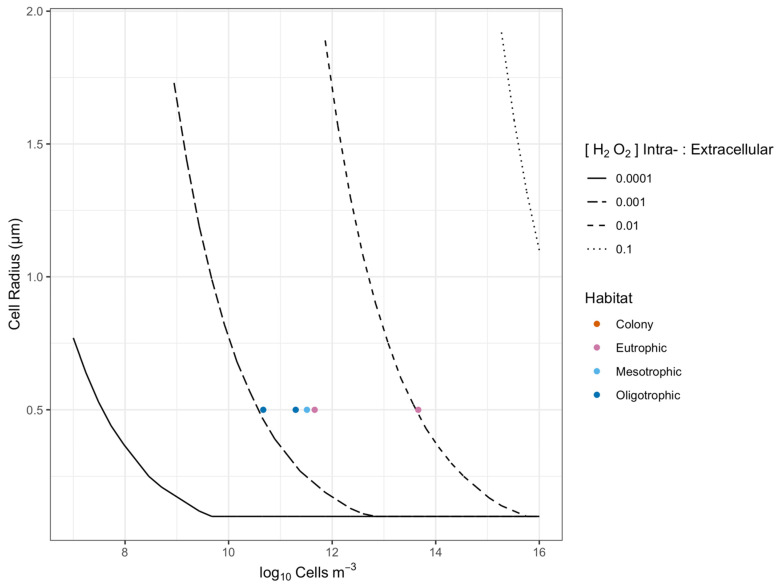
Thresholds for direct bacterioplankton cell-to-cell interaction through [H_2_O_2_] above seawater background. The X axis shows a range of cell m^−3^ (which in turn governs cell-to-cell distances), while the Y axis shows radii for cell-specific spheres of influence upon [H_2_O_2_] (a function of cell size, intracellular [H_2_O_2_] and seawater [H_2_O_2_]). The areas below the threshold lines show combinations where bacterioplankton cell-to-cell distance exceeds the cell-specific [H_2_O_2_] distance, so direct cell-to-cell interactions mediated by [H_2_O_2_] are unlikely. The area above the threshold lines show combinations where bacterioplankton cell-to-cell distance is less than the cell-specific [H_2_O_2_] distance, so direct cell-to-cell interactions mediated by [H_2_O_2_] are feasible. The four threshold lines refer to a range of ratios of bacterioplankton intracellular to extracellular [H_2_O_2_]. Data points from different habitats show that direct bacterioplankton cell-to-cell interactions are likely, over a range of reasonable ratios of intracellular to extracellular [H_2_O_2_], at >10^10^ cells m^−3^.

## Data Availability

The RStudio workbook used to generate this document is hosted at https://github.com/NaamanOmar/ROS_bioinfo.git.

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
