# Peer review of "Diffusional Interactions among Marine Phytoplankton and Bacterioplankton: Modelling H2O2 as a Case Study"

_microorganisms, 2022, doi:10.3390/microorganisms10040821_

Round 1
Reviewer 1 Report
Title
is misleading
Abstract
ok
Introduction
Table 1. There is one paper reporting a very high maximum number of single cells of cyanobium like cyanobacteria: 15 mio cells per ml as solitary cells (Klinkenberg G, Schumann R 1995 Abundance changes of autotrophic and heterotrophic picoplankton in the Zingster Strom, a shallow, tideless estuary south of the Darß-Zingst Peninsula (Southern Baltic Sea). Arch Hydrobiol 134:359-377). Or usually 3-5 mio cells per ml (Albrecht et al. 2017). Cell numbers in colonies are much higher. Refs 10 and 11 refer also to more marine environments. For chlorophytes and diatoms, you should look into limnetic water publications, where their cell numbers may be much higher than in brackish waters. I confess that it is not easy to extract those cell numbers as most authors give only biomass. If you are interested, I can provide cell numbers of the small diatom Stephanodiscus hantzschii from a very eutrophic river and a reference for biomass. All in all, I suggest to pay more attention to habitats salinity.
Lines 61 to 66. I do not understand and I do not see the reference. A figure or only a reference? It is hard for any reader to backtrack over all references used. Ref 18 refers only to small cyanobateria (derived from the title).
Methods
When I would have understood right from the beginning (title, abstract or introduction) that this paper is only about modelling, I would not have agreed to review. Now I am bound to this obligation and tried my best to interpret the physiological and ecological impacts of this study. However, this paper does not belong into Microorganisms.
What is the data basis for this modelling? You need to give it expicitely and add that to a data basis.
Results and discussion
First, I confess that I deeply dislike the structure of the papers format. I, however, acknowledge that it may be given by the journal. If a paper is concerned about modelling, I may suggest to look for another journal or indicate that somewhere highly on top of the manuscript. In such a paper a confusion of results and discussion is very hard for the readers, who have to differentiate between results (and these are model results) and interpretation.
Lines 153-160. What means highly expanded axes? Give examples, not only a warning, what may citing scientists overlook. Axes refer to cell numbers? I am confused.
Fig. 1 It is not clear to me, what the outcome should be. We have 2 point clouds, one more dense than the other.
Fig. 2 I do not understand why this different type of presentation is used. Data points indicate always a measuring point, which is not there. Stick to lines. It is not very helpful for an ecological reader.
Heading at line 207 Visualising …
As a microscoper, I expected something very different from plotting modelled results. This is unsatisfactory for me. The simplest way to come out would be to rephrase everything.
You explain a lot about cell dimensions, but what about cell chains of 1µm diameter and 20-100s of µm.
I did not follow discussion as it is mixed too much into results. So, I could not follow your results easily.
Author Response
Dear Reviewer,
Thank you for taking your time to review our paper.
Title is misleading
Thank you for pointing this out, we agree and have modified the title to clarify that the paper focuses on modelling H2O2 diffusional interactions and specifically focuses on Marine Phytoplankton and Bacterioplankton.
Table 1. There is one paper reporting a very high maximum number of single cells of cyanobium like cyanobacteria: 15 mio cells per ml as solitary cells (Klinkenberg G, Schumann R 1995 Abundance changes of autotrophic and heterotrophic picoplankton in the Zingster Strom, a shallow, tideless estuary south of the Darß-Zingst Peninsula (Southern Baltic Sea). Arch Hydrobiol 134:359-377). Or usually 3-5 mio cells per ml (Albrecht et al. 2017). Cell numbers in colonies are much higher. Refs 10 and 11 refer also to more marine environments. For chlorophytes and diatoms, you should look into limnetic water publications, where their cell numbers may be much higher than in brackish waters. I confess that it is not easy to extract those cell numbers as most authors give only biomass. If you are interested, I can provide cell numbers of the small diatom Stephanodiscus hantzschii from a very eutrophic river and a reference for biomass. All in all, I suggest to pay more attention to habitats salinity.
Thank you for the suggestion, we now cite data from Albrecht et al., 2017 and Klinkenberg and Schumann 1995 (Table 1). We have also clarified in the title that the manuscript focuses on marine phytoplankton. These changes are also reflected in Figure 5.
Lines 61 to 66. I do not understand and I do not see the reference. A figure or only a reference? It is hard for any reader to backtrack over all references used. Ref 18 refers only to small cyanobateria (derived from the title).
Thank you for catching this, we had missed a reference to Figure 1 at revised line 76.
When I would have understood right from the beginning (title, abstract or introduction) that this paper is only about modelling, I would not have agreed to review. Now I am bound to this obligation and tried my best to interpret the physiological and ecological impacts of this study. However, this paper does not belong into Microorganisms.
We apologize for the confusion and have changed the title to clarify that this invited paper focuses on modelling of H2O2 concentrations as outwards from cells and the potential functional implications of these patterns.
What is the data basis for this modelling? You need to give it expicitely and add that to a data basis.
We have added more on the data basis of the model (Revised lines 411 to 414) and have added more information on the literature supporting the assumptions of the model (Revised lines 111 to 122). Validating the assumptions of the simple model will be possible as more literature on phytoplankton reactive oxygen is published (Revised lines 380 to 382).
“All models are wrong, but some are useful.”
We hope that our simplified model may help provoke and guide efforts on diffusional exchange of reactive oxygen among phytoplankton.
Lines 153-160. What means highly expanded axes? Give examples, not only a warning, what may citing scientists overlook. Axes refer to cell numbers? I am confused.Fig. 1 It is not clear to me, what the outcome should be. We have 2 point clouds, one more dense than the other.
Revised lines 166 to 178. We have added additional information to the paragraph to improve the justification for the use of expanded axes.
Fig. 2 I do not understand why this different type of presentation is used. Data points indicate always a measuring point, which is not there. Stick to lines. It is not very helpful for an ecological reader.
Thank you for your comment.
We used black points to show the modelled concentrations of H2O2 under the influence of dilution alone moving out from the cell. We used red points to show the modelled concentrations of H2O2 under the combined influence of dilution and (negligible) degradation. The plotted lines simply connect the modelled points. We have revised the caption to clarify these usages.
As a microscoper, I expected something very different from plotting modelled results. This is unsatisfactory for me. The simplest way to come out would be to rephrase everything.
We hope the further explanations make this invited paper an easier read. As suggested by the reviewer, we have changed the title and emphasized that this is a modelling paper.
You explain a lot about cell dimensions, but what about cell chains of 1µm diameter and 20-100s of µm.
Thank you, this is an important consideration and would require more advanced modelling. We have included a note on this issue at revised lines 360 to 362.
I did not follow discussion as it is mixed too much into results. So, I could not follow your results easily.
We hope the further explanations make the revised paper an easier read. We also struggled to establish conceptual distinctions between modelled “results” and discussion.
Thank you again for taking the time to review our manuscript.
Sincerely,
Naaman Omar and Dr. Douglas A. Campbell.
On behalf of J. Scott P. McCain & Ondřej Prášil.
Reviewer 2 Report
The manuscript Microorganisms-1609030 addresses the influence of cells with high intracellular to extracellular H2O2 ratio (Phytoplankton according to the authors) and cells with a low ratio (Bacterioplankton, according to the authors) on H2O2 concentration in the surrounding environment, and the possible effects on neighboring cells.
The authors indicate the scarcity of data on H2O2 concentration on the cells (lines 102-105). They decided to use the value 1 uML-1 for phytoplankton cells, but it is unclear if this is representative. This data come from the reference given in line 170, but that review deals with mammalian cells. For bacterioplankton, they use an intracellular concentration of 1/10 of the surrounding medium. They don’t explain the origin of this value. They calculated the cell size, volume and concentration from bibliographic data. Considering a homogenous population of cells, they estimated the cellular distance. The central part of the manuscript is the estimation of the diffusion of H2O2 in the medium from the phytoplankton, or towards the bacterioplankton, and the distance to which those processes will affect the surrounding medium. The mathematical calculations used for the estimations would not be easily followed by most of the readers of the journal. There is no data that could allow testing if the predictions made are supported by any experimental observation.
The manuscript introduction is very nicely written and addresses a subject of scientific interest, as it is the estimation of the interaction between cells in terms of H2O2 production and degradation. The approach followed by the authors is interesting. In the manuscript submitted, they use a simple model, but they mention that they have the goal of developing more complex models taking into account additional variables. However, as the manuscript stands now, it lacks data supporting the model developed.
Author Response
Dear Reviewer,
Thank you for taking your time to review our paper.
The manuscript Microorganisms-1609030 addresses the influence of cells with high intracellular to extracellular H2O2 ratio (Phytoplankton according to the authors) and cells with a low ratio (Bacterioplankton, according to the authors) on H2O2 concentration in the surrounding environment, and the possible effects on neighboring cells.
The authors indicate the scarcity of data on H2O2 concentration on the cells (lines 102-105). They decided to use the value 1 uML-1 for phytoplankton cells, but it is unclear if this is representative. This data come from the reference given in line 170, but that review deals with mammalian cells. For bacterioplankton, they use an intracellular concentration of 1/10 of the surrounding medium. They don’t explain the origin of this value. They calculated the cell size, volume and concentration from bibliographic data. Considering a homogenous population of cells, they estimated the cellular distance. The central part of the manuscript is the estimation of the diffusion of H2O2 in the medium from the phytoplankton, or towards the bacterioplankton, and the distance to which those processes will affect the surrounding medium. The mathematical calculations used for the estimations would not be easily followed by most of the readers of the journal. There is no data that could allow testing if the predictions made are supported by any experimental observation.
Thank you for your comment. We have added more on the data basis of the model (Revised line 411 to 414) and have added more information on the literature supporting the assumptions of the model (Revised line 111 to 122). Validating the assumptions of the simple model will be possible as more literature on phytoplankton reactive oxygen is published (Revised line 380 to 382).
The manuscript introduction is very nicely written and addresses a subject of scientific interest, as it is the estimation of the interaction between cells in terms of H2O2 production and degradation. The approach followed by the authors is interesting. In the manuscript submitted, they use a simple model, but they mention that they have the goal of developing more complex models taking into account additional variables. However, as the manuscript stands now, it lacks data supporting the model developed.
We agree, and part of the motivation for writing this manuscript is the lack of necessary data in the field. We point out at revised lines 350 to 352 that the models are limited by the lack of published information on intracellular concentrations of H2O2 in phytoplankton.
Thank you again for taking the time to review our manuscript.
Sincerely,
Naaman Omar and Dr. Douglas A. Campbell.
On behalf of J. Scott P. McCain & Ondřej Prášil.
Round 2
Reviewer 1 Report
I understand, that the modelling results are focussing now to marine environments. But, why? Due to the data base you used? Or is there any difference in the models (diffusion) you used?
I still not get from the new version of the manuscript, what the modelled results tell us. Us means an ecologist. Please rewrite your conclusions (which are more a summary) into some outreach for a broader public.
I further miss a discussion about ROS as messengers. Maybe, there is a language gap between biological disciplines. There is, e.g. some outcome from messaging missing, cf. elke dittmann and refs there.
Author Response
Dear Reviewer,
Thank you again for taking your time to review our paper.
I understand, that the modelling results are focussing now to marine environments. But, why? Due to the data base you used? Or is there any difference in the models (diffusion) you used?
We focus on marine environments as the database that we used in our modelling focused on marine environments, as well as because we have estimates of normal [H2O2] in marine environments. We have further added a note that if cell densities and the [H2O2] in freshwater systems is similar to that of marine environments, our model would be comparable (Revised lines 337 to 341).
I still not get from the new version of the manuscript, what the modelled results tell us. Us means an ecologist. Please rewrite your conclusions (which are more a summary) into some outreach for a broader public.
Thank you for pointing this out, we have added more to our conclusions to further highlight that H2O2 diffusing out of cells likely does not directly interact with other cells (Revised lines 347 to 371)
I further miss a discussion about ROS as messengers. Maybe, there is a language gap between biological disciplines. There is, e.g. some outcome from messaging missing, cf. elke dittmann and refs there.
Thank you. We further expound on the importance of ROS in cell signaling and growth in revised lines 81 to 85 and 364 to 371.
Thank you again for taking your time to review our manuscript.
Sincerely,
Naaman Omar.
On behalf of J. Scott P. McCain, Ondřej Prášil and Douglas A. Campbell.
Reviewer 2 Report
The authors have provided new information in response to the comments. First, I have to mention that the lines with the new information indicated in their response are not correct…I assume that are lines 371-376, 100-107 and 340-342.
Besides, the authors indicate that the intracellular concentration in bacterial cells is 1/10 of the external value. However, in reference 49, this valued is provided for a experimental value, but is not mentioned in the text that this could be a fixed rule. In fact, alternative, it is mentioned that internal values will be 20 nM in the absence of extracellular hydrogen. I am not qualified to evaluate the mathematical model, but the starting values used in it are questionable.
In conclusion, this new version does not significantly improve the first submission.
Author Response
Dear Reviewer,
Thank you again for taking your time to review our paper.
The authors have provided new information in response to the comments. First, I have to mention that the lines with the new information indicated in their response are not correct…I assume that are lines 371-376, 100-107 and 340-342.
We apologize for the confusion, we had different line numbers in alternate versions of the manuscript.
Besides, the authors indicate that the intracellular concentration in bacterial cells is 1/10 of the external value. However, in reference 49, this valued is provided for a experimental value, but is not mentioned in the text that this could be a fixed rule. In fact, alternative, it is mentioned that internal values will be 20 nM in the absence of extracellular hydrogen. I am not qualified to evaluate the mathematical model, but the starting values used in it are questionable.
Thank you for your comment, due to limited intracellular concentrations from bacterioplankton cells, we had to use experimental values. We have noted that the models would need to be validated as more data becomes available in the end of the conclusion section (Revised lines 384 to 387). We have also addressed your point about the intracellular [H2O2] in revised lines 112 to 115 and 322 to 328.
In conclusion, this new version does not significantly improve the first submission.
Thank you for your comment, we have added more explanations to our conclusions to further reiterate the findings of our work (Revised lines 337 to 371). Unfortunately, we are limited by the lack of intracellular estimates of H2O2 and would have to validate the models as more data becomes available.
Thank you again for taking your time to review our manuscript.
Sincerely,
Naaman Omar.
On behalf of J. Scott P. McCain, Ondřej Prášil and Douglas A. Campbell.